# Incidence, predictors and re-treatment outcomes of recurrent myopic choroidal neo-vascularization

**Mukesh Jain**[1], **Raja Narayanan**[1,2]*, **Priya Jana**[3,4], **Ashik Mohamed**[4], **Rajiv Raman**[5], **Pavan Verkicharla**[4,6], **Srikanta Kumar Padhy**[1], **Anthony Vipin Das**[2,7], **Jay Chhablani**[8]

**1** Anant Bajaj Retina Institute, L V Prasad Eye Institute, Hyderabad, Telangana, India, **2** Indian Health Outcomes, Public Health, and Economics Research Centre (IHOPE), Hyderabad, Telangana, India, **3** Brien Holden Institute of Optometry and Vision Sciences, L V Prasad Eye Institute, Hyderabad, Telangana, India, **4** Ophthalmic Biophysics, L V Prasad Eye Institute, Hyderabad, Telangana, India, **5** Department of Vitreoretinal Services, Shri Bhagwan Mahavir Vitreoretinal Services, Sankara Nethralaya, Chennai, Tamil Nadu, India, **6** Myopia Research Lab, Prof. Brien Holden Eye Research Centre, L V Prasad Eye Institute, Hyderabad, Telangana, India, **7** Department of eyeSmart EMR & AEye, L V Prasad Eye Institute, Hyderabad, Telangana, India, **8** UPMC Eye Centre, University of Pittsburgh, Pittsburgh, PA, United States of America

* narayanan@lvpei.org

**Data Availability Statement:** Data cannot be shared publicly because of ethical and legal considerations. Data are available on reasonable request after clearance from the Hyderabad Eye

## Abstract

### Objectives

To evaluate incidence, predictors, and re-treatment outcome of recurrent myopic choroidal neovascularization (m-CNV).

### Methods

Retrospective consecutive observational series. From year 2014 to 2019, 167 eyes of 167 patients of treatment naïve m-CNV were enrolled. 59 and 108 eyes were treated with intra-vitreal ranibizumab and bevacizumab mono-therapy, respectively. Recurrence was defined as re-appearance of CNV activity, confirmed on optical coherence tomography (OCT) after at least 3 months of cessation of anti-VEGF therapy. Incidence of recurrence, predictors and re-treatment outcomes were studied.

### Results

Overall, mean age and spherical equivalence (SE) was 47.95 ± 14.72 years and -12.19 ± 4.93 D respectively. Males constituted 50.9%. 44 eyes (26.4%) had a recurrence during a mean follow up of 16.5 ± 12.86 months. Kaplan-Meier survival analysis showed the risk of recurrence was 8, 26 and, 33.6% at 6, 12 and 18 months, respectively. Age (p = 0.511), gender (p = 0.218), SE (p = 0.092), anti-VEGF (p = 0.629) and baseline BCVA (p = 0.519) did not influence recurrence. Number of injections administered to control the disease in the first episode was the only significant predictor of recurrence (Cox Proportional Hazard Ratio 2.89–3.07, 95% Confidence Interval: 1.28–7.45; p = 0.005). At 12 months, eyes requiring one injection in first episode had a recurrence rate of 12% versus 45% in eyes requiring 3 or more injections in the first episode. A mean number of 1.9 additional injections per eye was

Institute Ethics Committee (contact via Mrs Charitha Thiruttani, charitha@lvpei.org) for researchers who meet the criteria for access to confidential data.

**Funding:** RAJA NARAYANAN DBT Wellcome Trust India Alliance Clinical Research Centre Grant IA/CRC/19/1/610010 NO- The funders had no role in study design, data collection and analysis, decision to publish, or preparation of the manuscript.

**Competing interests:** NO AUTHORS HAVE ANY COMPETING INTEREST

**Abbreviations:** IVR, Intra-vitreal ranibizumab; IVB, Intra-vitreal bevacizumab; m-CNV, myopic choroidal neovascularization; VEGF, vascular endothelial growth factor; LogMAR, logarithm of the minimum angle of resolution; OCT, optical coherence tomography; SE, spherical equivalence.

needed during re-treatment. Final BCVA in the recurrence group was similar to that of non-recurrence group (0.53 ± 0.40 versus 0.55 ± 0.36 LogMAR; p = 0.755). Baseline BCVA (p = 0.0001) was the only predictor of final visual outcome irrespective of anti-VEGF drug (p = 0.38).

## Conclusion

Eyes requiring greater number of injections for disease control in first episode are "at risk" of early m-CNV recurrence. However, recurrence does not adversely affect visual outcome, if treated adequately.

## Introduction

The incidence of myopia is increasing worldwide [1,2]. Myopic choroidal neo-vascularization (m-CNV) is a common vision threatening complication in high myopic eyes. m-CNV is estimated to occur in 4%-10% of patients with pathological myopia, and if left untreated, can cause rapid vision loss [3–6]. In particular, m-CNV affects middle-aged individuals resulting in additional significant economic, social, and emotional burdens [3–6].

Multiple treatment modalities have been tried in the past. However, long-term outcomes were poor and associated with high recurrence rates. Anti-vascular endothelial growth factors (anti-VEGFs) are the standard of care today [7–16]. Bevacizumab is a cost-effective alternative to ranibizumab and aflibercept in the treatment of m-CNV [17–27]. Unlike age-related macular degeneration CNV, there is a paucity of literature on incidence, possible predictors and re-treatment outcomes in recurrent m-CNV [28–34]. This information is crucial in optimizing follow-up regimen of m-CNV eyes to detect early re-activation and counselling patients about re-treatment outcomes.

In this study, we reviewed our large data comprising of 167 eyes of 167 patients diagnosed with naïve m-CNV treated with either of the two anti-VEGFs, namely ranibizumab and bevacizumab. We primarily aimed to evaluated the incidence, possible predictors and re-treatment outcomes. Secondarily, e also compare the efficacy and safety of these two anti-VEGF drugs in the treatment of m-CNV.

## Materials and methods

The retrospective study approval was taken from the Institutional Ethics Committee [LEC-BHR-R-11-20549] and the study adhered to the tenets of the Declaration of Helsinki. Electronic medical records were searched to identify treatment naïve cases of "Myopic CNV" which were treated with anti-VEGF injections monotherapy during the study period of January 2014 to December 2019. A standard consent form for electronic data privacy and consent for the use of data for research purpose was filled by the patients at the time of registration. No identifiable information of the patient was used for analytical purposes.

Medical records of consecutive cases fulfilling the inclusion criteria were reviewed. Key eligibility criteria included: 1. patient age ≥18 years; 2. myopia <-6.00 D; 3. sub-foveal and juxta-foveal location of m-CNV; 4. presence of m-CNV confirmed on multimodal imaging; 6. best-corrected visual acuity (BCVA) of 20/30 to 20/400 in the study eye.

Key exclusion criteria included: (1) previous vitreoretinal surgery in the study eye; (2) previous macular laser photocoagulation or photodynamic therapy in the study eye; (3) decrease in

visual acuity due to causes other than m-CNV; (4) switching anti-VEGF drugs during the treatment period; (5) follow-up of less than 6 months after cessation of anti-VEGF therapy. We also excluded eyes with other disorders known to be associated with choroidal neovascularization and/or macular edema such as age-related macular degeneration, inflammation, angioid streaks, trauma, dystrophies, diabetic retinopathy and retinal vein occlusion. We excluded cases that had insufficient clinical and investigation details or with diagnostic dilemmas.

All patients underwent a detailed ocular evaluation including BCVA testing, dilated fundus examination with slit-lamp bio-microscopy, and multimodal imaging, as needed. Details related to the two anti-VEGF agents were explained to help in the informed independent decision by the patients. Intravitreal injection of ranibizumab (0.5 mg/0.05 ml) or bevacizumab (1.25 mg/0.05ml) was given under aseptic precautions. After the first anti-VEGF injection, re-treatment was continued monthly using disease activity criteria. Eyes with switching of anti-VEGF have been excluded from this present study.

Recurrence was defined as reappearance of m-CNV activity after cessation of anti-VEGF therapy for at least 3 months: [1] drop in BCVA as compared to the last visit and/or new onset metamorphopsia; [2] new cystoid/sub-retinal fluid and/or definitive blurring, fuzziness and increase in the size of CNV lesion margins on OCT as compared to the last scan.

Data collected and tabulated included age, gender, spherical equivalent, pre-injection BCVA, anti-VEGF injection used, and number of injections administered, duration of follow-up, recurrence if any, and final BCVA. BCVA was assessed using the Snellen's chart listed as the logarithm of the minimum angle of resolution (LogMAR) equivalents for statistical analysis.

The statistical analysis was performed using STATA v14.2 (StataCorp, College Station, TX, USA). The distribution of continuous data were checked for normality by Shapiro-Wilk test. Summary measures included mean with standard deviation and proportions for continuous and categorical data, respectively. Kaplan-Meier survival analysis was used to estimate the probability of recurrence. The equality of survivor functions was assessed by log-rank test. Cox-proportion hazard regression analysis was done to find the predictors of recurrence and 95% confidence level of hazard ratio. A p-value of <0.05 was considered statistically significant.

Secondary analysis to compare non-inferiority of IVB to IVR was done. Overall BCVA of all study subjects among different visits, a mixed-effects model with random intercept at the subject level was used and marginal linear predictions were compared. Comparisons between study groups were performed by Mann-Whitney test and Chi-square test for continuous and categorical data respectively. A bivariate Spearman correlation analysis was used to evaluate the factors associated with post-injection BCVA, followed by a multiple regression analysis to find the significant predictors of good visual outcome after treatment. In multiple pairwise comparison, a p-value of <0.025 was considered statistically significant after adjustment for Bonferroni correction.

## Results

During the period January 2014 to December 2019, 167 eyes of 167 patients were included in the study. Overall, mean age and spherical equivalence (SE) was 47.95 ± 14.72 years and -12.19 ± 4.93 D respectively. Males constituted 50.9%. Mean presenting BCVA was 0.68 +/- 0.38 LogMAR. After an initial mean number of 2.33 ±1.46 injections per eye for initial disease control, mean BCVA improved to 0.51 ± 0.34 LogMAR (p< 0.05). Vision remained stable (p = 0.10) over a mean follow-up of 16.55 ± 12.86 months, 0.55 ± 0.37 LogMAR at final visit.

**Table 1. Shows the Cox-proportional hazards regression analysis to determine the predictors of recurrence.**

| Variable | Sub-group | p value | Hazard Ratio | 95% CI for HR | |
|---|---|---|---|---|---|
| | | | | 5% | 95% |
| **Gender** | **Male (50.9%)** | | 1 | | |
| | **Female (49.1%)** | 0.218 | 1.522 | 0.80 | 2.96 |
| **Age** | **Less than 40 years (33.3%)** | | 1 | | |
| | **40–60 years (34.1%)** | 0.511 | 1.2 | 0.53 | 2.78 |
| | **More than 60 years (32.3%)** | | .75 | 0.36 | 1.58 |
| **S.E** | **More than -9 D (36.8%)** | | 1 | | |
| | **-9 to -14 D (32.3%)** | 0.092 | 1.03 | 0.37 | 2.90 |
| | **Less than -14 D (30.8%)** | | 0.42 | 0.15 | 1.12 |
| **Anti-VEGF** | **Bevacizumab (64.7%)** | | 1 | | |
| | **Ranibizumab (35.3%)** | 0.629 | 1.17 | 0.61 | 2.27 |
| **Baseline BCVA** | **0.2–0.4 LogMAR (29.9%)** | | 1 | | |
| | **0.5–0.8 LogMAR (40.7%)** | 0.519 | 0.71 | 0.32 | 1.58 |
| | **>0.8 LogMAR (29.3%)** | | 0.79 | 0.45 | 2.43 |
| **No of injections** | **1 injection(30.5%)** | | 1 | | |
| | **2 injections(36.5%)** | **0.005** | **3.07** | **1.28** | **7.40** |
| | **3 or more injections(32.9%)** | | **2.89** | **1.35** | **6.45** |

Percentage of eyes with BCVA >20/50 (Snellen equivalent) was 43.7% in the final visit as compared 29.9% at presentation.

Of the total 167 eyes, 44 (26.4%) eyes had recurrence. Cox-proportional hazard regression analysis was done to determine the predictors of recurrence (Table 1). Gender (p = 0.218), age at presentation (p = 0.51), SE (p = 0.092), anti-VEGF used (p = 0.6385) and baseline BCVA (p = 0.52) were not associated with the probability of recurrence (Table 1). Only the number of injections administered to control the disease activity in the first episode had a significant effect (p = 0.005; Hazard ratio 2.89–3.07, 95% Confidence Interval 1.28–7.4) (Table 1).

Kaplan Meier Survival analysis was done to calculate the risk of recurrence (Table 2). Overall, Kaplan-Meier survival analysis showed the risk of recurrence was 8, 26 and 34% at 6, 12 and 18 months, respectively (Table 2). At 12 months, the risk of recurrence was 12.1, 15.0 and

**Table 2. Shows Kaplan Meier Survival table analysis overall, anti-VEGF, and number of injections groups.**

| Time (months) | All eyes | | Bevacizumab | | Ranibizumab | |
|---|---|---|---|---|---|---|
| | Number at risk | Recurrence ± SE | Number at risk | Recurrence ± SE | Number at risk | Recurrence ± SE |
| 3 | 165 | 1.8% ± 1.0% | 107 | 0.9% ± 0.9% | 58 | 3.4% ± 2.4% |
| 6 | 153 | 7.8% ± 2.1% | 99 | 7.5% ± 2.5% | 54 | 8.5% ± 3.6% |
| 12 | 67 | 25.9% ±4.1% | 36 | 29.1% ± 5.7% | 31 | 22.0% ± 6.0% |
| 18 | 39 | 33.6% ± 4.9% | 18 | 38.5% ± 7.1% | 21 | 28.1% ± 6.9% |
| | One Injection | | 2 Injections | | 3 Injections or more | |
| 3 | 50 | 3.9% ± 2.7% | 61 | 0.0% | 55 | 1.8% ± 1.8% |
| 6 | 47 | 7.9% ± 3.8% | 57 | 5.0% ± 2.8% | 49 | 10.9% ± 4.2% |
| 12 | 22 | 12.1% ± 5.5% | 23 | 15.0% ± 6.1% | 22 | 45.5% ± 7.5% |
| 18 | 15 | 12.1% ± 5.5% | 12 | 31.1% ± 9.7% | 13 | 53.3% ± 8.2% |

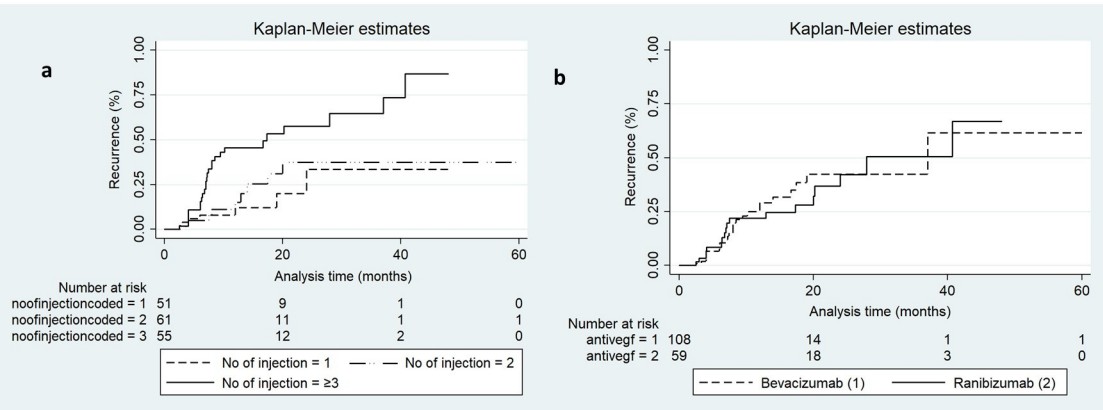

**Fig 1.** (a) Shows the survival function curve for the number of injections, depicting a steady increase in recurrence rate with the number of injections. (b) Shows the survival function curve for the bevacizumab and ranibizumab, depicting no difference in the recurrence rate between the two groups.

45.5% in the one injection, two injection and 3 or more injections groups, respectively (Fig 1A). At 18 months, the risk of recurrence was 12.1 (stable), 31.1 (two-fold increase) and 53.3% (marginal increase) in the one injection, two injection and 3 or more injections groups, respectively. The recurrence rate was similar in bevacizumab and ranibizumab group (Fig 1B).

Response to re-treatment was evaluated in 44 eyes with recurrent m-CNV. Final BCVA in the recurrence group was not statistically different from that in non-recurrence group (0.53 ± 0.40 versus 0.55 ± 0.36 LogMAR; p = 0.755). A mean number of additional 1.90 ± 1.78 injection per eye was needed to treat recurrence.

A secondary analysis was done to compare the efficacy and treatment burden between IVR and IVB groups and predictors of favourable visual outcomes.

A sub-group analysis of the two anti-VEGF subgroups showed no difference in the mean age, gender, mean spherical equivalence and mean number of injections (Table 3). Mean

**Table 3. Shows the baseline characteristics and visual outcomes in the overall, bevacizumab and ranibizumab group.**

| Characteristics | | Overall (N = 167 eyes) | Bevacizumab (N = 108 eyes) | Ranibizumab (N = 59 eyes) | P-value |
|---|---|---|---|---|---|
| Age (years), mean ± SD | | 47.95 ± 14.72 | 47.77 ± 14.10 | 48.27 ± 15.91 | 0.91 |
| Males, n (%) | | 85 (50.9%) | 56 (51.9%) | 29 (49.2%) | 0.86[#] |
| Spherical equivalence (D), (mean ± SD) | | 12.19 ± 4.93 | 12.02 ± 5.06 | 12.47 ± 4.71 | 0.49 |
| Number of injections, (mean ± SD) | | 2.33 ± 1.46 | 2.32 ± 1.52 | 2.34 ±1.35 | 0.62 |
| Follow-up (months), (mean ± SD) | | 16.55 ± 12.86 | 15.33 ± 12.98 | 18.76 ± 12.43 | 0.02 |
| BCVA (LogMAR) (mean ± SD) | Pre-injection | **0.68 ± 0.38[$]** | **0.72 ± 0.34[$]** | **0.62 ± 0.43[$]** | 0.02 |
| | Post-Injection | **0.51 ± 0.34[$]** | **0.57 ± 0.34[$]** | **0.40 ± 0.32[$]** | 0.007[a] |
| | Gain in BCVA | 0.17 ± 0.29 | 0.14 ± 0.26 | 0.22 ± 0.34 | 0.006[b] |
| | Final follow-up | 0.55 ± 0.37 | 0.61 ±0.35 | 0.44 ± 0.37 | 0.03[c] |

SD: Standard Deviation; D: Dioptre; n = number; BCVA- Best Corrected Visual Acuity.

$: Paired t test is significant with p = 0.00

[#] chi-quare test

[a]adjusted for pre-injection BCVA (co-efficient: -0.11 ± 0.04)

[b]adjusted for pre-injection BCVA (co-efficient: 0.12 ± 0.04)

[c]adjusted for pre-injection BCVA and follow-up duration (co-efficient: -0.12 ± 0.05).

Table 4. Shows the multiple linear regression for post-injection BCVA as the dependent variable.

| Variable | Spearman correlation coefficient | P value | Multiple regression | |
|---|---|---|---|---|
| | | | p-value | Co-efficient |
| Age | 0.17 | 0.03 | 0.09 | - |
| SE | -0.07 | 0.49 | 0.78 | - |
| Baseline BCVA | 0.76 | <0.0001 | **<0.0001** | 0.82 ± 0.06 |
| Gender | -0.06 | 0.44 | 0.83 | - |
| Anti-VEGF | -0.24 | 0.002 | 0.38 | - |
| No. of Injections | -0.15 | 0.049 | 0.09 | - |

SE: Spherical equivalent; BCVA: Best correct visual acuity; Anti-VEGF: Anti-vascular endothelial growth factor; No.: Number.

presenting BCVA in the ranibizumab sub-group was 0.62 ± 0.43 LogMAR which improved by a mean of 0.22 ± 0.34 LogMAR at the end of intravitreal injection for initial disease control to 0.40 ± 0.32 LogMAR. In comparison, in the bevacizumab sub-group the mean presenting BCVA (0.72 ± 0.34 LogMAR, p = 0.02), gain in BCVA (0.14 ± 0.26 LogMAR, p = 0.006) and BCVA (0.57 ± 0.34 LogMAR, p = 0.03) was lower. This could possibly reflect delay in seeking medical attention because of lower socio-economic status of IVB group in real world settings.

Bivariate Spearman correlation analysis showed age, baseline BCVA, type of anti-VEGF and number of injections to be significantly associated with post-injection BCVA (Table 4). Multiple regression analysis, by backward elimination, was then performed (Table 4). We found that presenting visual acuity (p<0.0001) was the only factor that correlated with post-injection BCVA irrespective of the anti-VEGF drug (p = 0.38) and the number of injections administered (p = 0.09) (Table 4). Therefore, IVB is non-inferior to IVR with no additional treatment burden in m-CNVM.

The adverse events noted in the follow-up period were: intraocular pressure rise: none; retinal detachment 2 cases (1 each in IVR and IVB) and endophthalmitis 1 case (IVB group).

## Discussion

In this study, we found nearly 1/4th of the eyes with m-CNV had recurrence on follow-up which responded favourably with repeat anti-VEGF treatment. Only predictor of recurrence was the number of injections needed to control the initial disease activity. Eyes requiring 3 or more injection for initial disease stabilization are particularly "at risk" for early recurrence. IVB is non-inferior to IVR; good baseline vision was the only predictor of favourable visual outcome after anti-VEGF treatment.

We found the mean age of presentation was 47.9 years which is almost a decade earlier as compared to 56.1 and 58.5 years in RADIANCE and MYRROR (Asian population) studies [14,15]. A study by Rishi et al from India has reported a similar earlier presentation as observed in our series [27]. Genetic factors may result in this difference observed in the Indian sub-continent. Early presentation in their productive years has significant emotional, financial, and social-economic implications which must be borne in mind when treating these patients. As compared to female pre-ponderance in RADIANCE and MYRROR studies [14,15], we found no gender predisposition as that observed by Rishi et al. [27].

Limited literature exists on the incidence, predictors and re-treatment outcomes of m-CNV [32–35]. The reported recurrence rate in m-CNV is variable. Yang et al. (103 eyes), Siu-Chang et al. (93 eyes), Kang et al. (76 eyes) and, Jo et al. (58 eyes) reported 23.3%, 26.9%, 46.1% and 34% recurrence rate in their retrospective series of m-CNV [28–30,32]. In RADIANCE study, of 116 eyes being treated with ranibizumab injections based on disease activity, 37.1% of eyes

required repeat injection between 6 and 11 months [15]. In the present series we found 44 eyes (26.3%) had recurrences over a mean follow up of 16.5 months. Variability in reported recurrence rate across series results from non-uniformity in "recurrence" criteria, follow-up regimen and duration, availability of OCT/multi-modal imaging, mode of treatment (PDT/anti-VEGF/mixed) and protocols followed.

Previous studies unequivocally suggest that most recurrences occur early, especially during the first year. In retrospective series by Wang et al., 72.7% of all recurrence occurred in the first year of treatment [30]. Similarly, Tan et al. in post-RADIANCE observation study reported a recurrence rate of only 10% between 12–48 months versus 37.1% between 6 to 11 months [31]. In accordance, we found risk of recurrence was 8% at 6 months, which increased significantly to 26% at 12 months follow-up. Thereafter, it increased only marginally to 34% at 18 months. These findings of ours reinforce the importance of careful monitoring for recurrence in the first year after cessation of anti-VEGF treatment in m-CNV eyes.

Since it is difficult to predict recurrence, knowledge of possible predictors can help to identify "at risk" eyes. Various morphological and treatment modalities have been suggested as possible predictors of recurrence [28–34]. Kang et al. found that use of PDT as treatment modality resulted in more recurrence [29]. Jo et al. found that greater the number of injections needed for initial disease control, the higher was the risk of recurrence [28]. In this present study we also found that the number of injections needed for initial disease control significantly predicted recurrence. A significant proportion of eyes with early recurrence belonged to the sub-group requiring 3 or more injections to control the disease activity in the first episode [Fig 1A]. This is an important information of our study suggesting these eyes are "at risk" and must be followed meticulously with repeated OCT. We found similar recurrence rate between bevacizumab and ranibizumab group, which further validates the non-inferior of bevacizumab [Fig 1B].

Although the risk of recurrence was high, we found that m-CNV recurrence respond well to re-treatment. The final BCVA in the recurrence group was similar to that in the non-recurrence group with an additional 1.90 injection per eye needed to treat recurrence. This information will be helpful in counselling patients with recurrence about the good prognosis despite recurrence with continued anti-VEGF treatment.

The cost and logistics are a major consideration when advising anti-VEGF treatment in real world. Since our study represent real world experience, the mean presenting BCVA in our study was lower with wider variation (0.68 ± 0.38LogMAR, Snellen's equivalent 20/100) as compared to RADIANCE (equivalent ETDRS letters 55 ± 12, Snellen's equivalent 20/80) and MYRROR studies (equivalent ETDRS letters 56 ± 9.6, Snellen's equivalent 20/80) [15,16]. The common belief of suboptimal outcome in real world scenario may be deceptive. Poorer baseline visual acuity in real world setting may be the real confounding factor, as supported by our rigorous regression analysis. We found bevacizumab is non-inferior to ranibizumab with no additional treatment burden.

In accordance with all the previous studies, we also found that the gain in visual outcome with anti-VEGF injections was maintained over long follow-up [35,36].

The strength of the study is single-centre real world settings with reasonably large sample size and robust statistical analysis. Significant attrition during long term follow-up in real world settings prevented estimation of recurrence rate beyond 24 months. However, recurrences in m-CNV beyond this period seems to increase only marginally, and are of little clinical significance. Lack of quantifiable OCT angiograph and fluorescein angiography measurements and repeated axial length measurents are limitation of this retrospective study. A large future prospective study would help to unravel the present lacunae in our understanding

In conclusion, we found the risk of recurrence in m-CNV is high. Number of injections needed to stabilize initial disease activity predicted recurrence. Eyes requiring 3 or more injections for initial stabilization specifically are "at risk" of early recurrence. However, recurrence does not alter the final visual outcome if treated appropriately. Further prospective studies are needed to validate our results.

## Acknowledgments

We would like to thank Mr. Pasha, Electronic Medical Record Team at LVPEI, Hyderabad for helping us.

## Author Contributions

**Conceptualization:** Raja Narayanan.

**Data curation:** Mukesh Jain, Priya Jana, Srikanta Kumar Padhy.

**Formal analysis:** Mukesh Jain, Priya Jana, Ashik Mohamed.

**Funding acquisition:** Raja Narayanan, Ashik Mohamed.

**Investigation:** Priya Jana, Srikanta Kumar Padhy.

**Methodology:** Mukesh Jain, Priya Jana, Ashik Mohamed, Pavan Verkicharla, Srikanta Kumar Padhy, Anthony Vipin Das.

**Project administration:** Raja Narayanan.

**Resources:** Raja Narayanan, Pavan Verkicharla.

**Software:** Anthony Vipin Das.

**Supervision:** Mukesh Jain, Raja Narayanan, Rajiv Raman, Anthony Vipin Das.

**Validation:** Mukesh Jain, Jay Chhablani.

**Writing – review & editing:** Mukesh Jain, Raja Narayanan, Rajiv Raman, Pavan Verkicharla, Anthony Vipin Das, Jay Chhablani.

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
