## [Decision Letter · Decision Letter 0]

22 Feb 2022

PONE-D-21-34518Incidence, Predictors and Re-treatment Outcomes Of Recurrent Myopic Choroidal Neo-vascularizationPLOS ONE

Dear Dr. Narayanan,

Thank you for submitting your manuscript to PLOS ONE. After careful consideration, we feel that it has merit but does not fully meet PLOS ONE’s publication criteria as it currently stands. Therefore, we invite you to submit a revised version of the manuscript that addresses the points raised during the review process

We look forward to receiving your revised manuscript.

Kind regards,

Ramesh Venkatesh

Academic Editor

PLOS ONE

Journal Requirements:

Reviewers' comments:

Reviewer's Responses to Questions

**Comments to the Author**

1. Is the manuscript technically sound, and do the data support the conclusions?

Reviewer #1: Yes

2. Has the statistical analysis been performed appropriately and rigorously? 

Reviewer #1: Yes

3. Have the authors made all data underlying the findings in their manuscript fully available?

Reviewer #1: Yes

4. Is the manuscript presented in an intelligible fashion and written in standard English?

Reviewer #1: Yes

5. Review Comments to the Author

Reviewer #1: Nice study which looks at the possible factors affecting recurrence in myopic CNV after Anti VEGF treatment. The only factor influencing recurrence was the number of injections to control the disease activity in the first episode. Have a few queries.

1. Was axial length calculated at each visit, the continued increase in the axial length over a period of time could be one factor for the recurrence.

2.Was FFA done before cessation of the Anti VEGF treatment?. Without FFA it would be difficult to say that the membrane was inactive.

3.Quantitative profile of the mCNV could have given us more information. Did any patient undergo OCTA?

4. Did the area of mCNV on FFA have a role to play in the recurrence?

6. PLOS authors have the option to publish the peer review history of their article (what does this mean?). If published, this will include your full peer review and any attached files.

Reviewer #1: No

---

## [Author Response · Author response to Decision Letter 0]

27 May 2022

Thank you for your valuable comments. We have included the changes as suggested. Thank you again.

---

## [Editor Report · Decision Letter 1]

29 Jun 2022

Incidence, Predictors and Re-treatment Outcomes Of Recurrent Myopic Choroidal Neo-vascularization

PONE-D-21-34518R1

Dear Dr. Narayanan,

We’re pleased to inform you that your manuscript has been judged scientifically suitable for publication and will be formally accepted for publication once it meets all outstanding technical requirements.

Kind regards,

Ramesh Venkatesh

Academic Editor

PLOS ONE

Additional Editor Comments (optional):

The reviewers comments have been addressed. Thanks for the opportunity.
---

## [Editor Report · Acceptance letter]

14 Jul 2022

PONE-D-21-34518R1 

Incidence, Predictors and Re-treatment Outcomes Of Recurrent Myopic Choroidal Neo-vascularization 

Dear Dr. Narayanan:

I'm pleased to inform you that your manuscript has been deemed suitable for publication in PLOS ONE. Congratulations! Your manuscript is now with our production department. 

Kind regards, 

on behalf of

Dr. Ramesh Venkatesh 

Academic Editor

PLOS ONE